# Distance Matters: A Distance-Aware Medical Image Segmentation Algorithm

**DOI:** 10.3390/e25081169

**Published:** 2023-08-05

**Authors:** Yuncong Feng, Yeming Cong, Shuaijie Xing, Hairui Wang, Cuixing Zhao, Xiaoli Zhang, Qingan Yao

**Affiliations:** 1College of Computer Science and Engineering, Changchun University of Technology, Changchun 130012, China; fengyuncong@ccut.edu.cn (Y.F.); 2202103018@stu.ccut.edu.cn (Y.C.); 2202103102@stu.ccut.edu.cn (S.X.); 2202103038@stu.ccut.edu.cn (H.W.); 2202103110@stu.ccut.edu.cn (C.Z.); yao@ccut.edu.cn (Q.Y.); 2Artificial Intelligence Research Institute, Changchun University of Technology, Changchun 130012, China; 3Key Laboratory of Symbolic Computation and Knowledge Engineering of Ministry of Education, Jilin University, Changchun 130012, China

**Keywords:** medical image segmentation, global context vision transformer, attention, TransUnet, global token generator

## Abstract

The transformer-based U-Net network structure has gained popularity in the field of medical image segmentation. However, most networks overlook the impact of the distance between each patch on the encoding process. This paper proposes a novel GC-TransUnet for medical image segmentation. The key innovation is that it takes into account the relationships between patch blocks based on their distances, optimizing the encoding process in traditional transformer networks. This optimization results in improved encoding efficiency and reduced computational costs. Moreover, the proposed GC-TransUnet is combined with U-Net to accomplish the segmentation task. In the encoder part, the traditional vision transformer is replaced by the global context vision transformer (GC-VIT), eliminating the need for the CNN network while retaining skip connections for subsequent decoders. Experimental results demonstrate that the proposed algorithm achieves superior segmentation results compared to other algorithms when applied to medical images.

## 1. Introduction

Medical image segmentation is a crucial technique used in the field of medicine. It involves dividing medical images into distinct regions based on specific similarity features. This segmentation process helps identify and isolate features that hold significant clinical relevance [1,2,3]. By providing a reliable foundation for clinical diagnosis, treatment, and pathological research, it empowers medical professionals to make more accurate and informed diagnoses. However, medical images frequently encounter obstacles, such as low contrast and a low signal-to-noise ratio, as well as organ movement, deformation, and variations among individuals depicted in the images. These challenges substantially amplify the difficulty and complexity associated with medical image segmentation [4,5,6].

Nowadays, medical image segmentation algorithms based on convolutional neural networks (CNNs) have become mainstream [7,8,9,10]. These algorithms include fully convolutional neural networks (FCNs), U-Net, and U-shaped network structures [11,12,13]. By continuously extracting features and utilizing the encoder–decoder structure, CNN algorithms can effectively recognize and segment important regions in medical images [14,15,16]. However, due to its inherent induction bias constraint, the CNN is unable to complete global modeling and obtain comprehensive global information [17,18]. As a result, the segmented areas may have blurred contours, and determining boundary information may be difficult.

Thanks to the remarkable success of transformers in natural language processing, vision transformers are increasingly being utilized as backbone networks in medical image segmentation [19,20,21]. Dosovitskiy A et al. introduced the concept of visual transformers (VITs) [22]. In the VIT, an image is first divided into several smaller patch blocks, with each block typically measuring 16 × 16 pixels. Subsequently, each patch block is projected into a fixed-length vector and passed through a transformer network. Concurrently, token and location codes are incorporated into the process. Through this sequence of operations, the VIT has demonstrated impressive accuracy and robustness in image classification tasks. Subsequently, the transformer architecture began to find applications in computer vision tasks, with the VIT serving as a significant milestone in this domain. In an effort to mitigate the computational complexity of the VIT and optimize its parameters, Touvron et al. introduced DeiT [23]. DeiT not only simplifies the training process of the VIT but also enhances the performance of image classification by leveraging knowledge distillation techniques. This advancement in the form of DeiT offers a promising avenue for improving the efficiency and effectiveness of transformer-based image classification models.

With the growing applicability of transformers in medical image segmentation, researchers have increasingly integrated transformers and CNNs to accomplish segmentation tasks [24,25]. Prominent examples of such networks include TransUnet [26] and LeVit-Unet [27]. TransUnet was the pioneering work that successfully combined these two approaches. In TransUnet, a VIT is employed for the encoder part, while U-Net is used for the decoder part. Additionally, three skip connections are incorporated. This network demonstrates improved accuracy and robustness compared to traditional models. Following the success of TransUnet, a series of transformer-based methods have emerged, including DS-TransUNet [28], AFTer-UNet [29], MT-UNet [30], and FCN-Transformer [31]. In these networks, the combination of convolutional neural networks (CNNs) and transformers empowers the network to simultaneously extract detailed features and perform global modeling. This effectively addresses the limitations of CNNs in segmenting image edge information. Nevertheless, both TransUent and LeVit-Unet encounter challenges in encoding and interacting with multiscale features. Additionally, due to the inherent encoding characteristics of vision transformers, they are unable to establish strong connections based on the distance between each patch block. These networks uniformly encode all patch blocks, resulting in a reduced encoding efficiency and increased overhead.

To address the aforementioned challenges, we propose a novel approach for medical image segmentation named GC-TransUnet. Our algorithm combines a CNN and transformer architecture, leveraging an encoder–decoder structure. To ensure efficient transformer encoding, we incorporate local attention, global attention, and global token generator (GTG) modules. These modules enable distinguishing and encoding patch blocks at varying distances during the encoding process.In the first step, the image to be segmented is fed into the local attention module, where each patch block and its neighboring blocks undergo self-attention operations, facilitating local encoding. Next, we utilize the GTG module to extract crucial features from the image, allowing important information from patch blocks situated further away from the central patch block to be aggregated within a feature map. Finally, the global attention module performs operations between each central patch block and the information contained in the feature map. This interaction bridges the gap between the central patch block and those located at longer distances. By employing this encoding scheme, we maximize the extraction of important information across distant patch blocks, consequently eliminating irrelevant information and mitigating the impact of noise. Ultimately, this approach enhances encoding efficiency. Furthermore, we incorporate the patch expand module [32,33] to facilitate the necessary adjustments in image size and channels, ensuring compatibility with the decoder. During the decoding stage, we employ the U-Net architecture to carry out the decoding process. Additionally, we perform four rounds of encoding to obtain four feature maps of different sizes, thus capturing multiscale information. The contributions of this paper can be summarized as follows:

(1) The proposed algorithm incorporates the GTG module, effectively filtering long-distance information. Additionally, we utilize a pyramid structure to progressively downscale the feature map, capturing long-distance information at different scales.

(2) To ensure efficient encoding, we employ both local attention and global attention mechanisms. These mechanisms enable the interactive encoding of long-range information at varying scales, thus enhancing coding efficiency.

(3) By combining a transformer with U-Net, our model integrates both detailed information and long-range information, allowing for a comprehensive representation of the input data.

The rest of this paper is organized as follows: In Section 2, we describe the motivation of the proposed algorithm. Section 3 details our proposed algorithm. Section 4 depicts segmentation results and analysis on different medical images. Conclusions and future work are presented and discussed in Section 5.

## 2. Motivation

The correlations between each pixel block in an image can vary significantly. For instance, adjacent pixel blocks may share similar features, while those that are far apart usually have fewer connections. Traditional transformers tend to overlook the correlation between pixel blocks or introduce additional operations like interactive windows, which can lead to unnecessary computational resource wastage.

In Figure 1, an example is presented to illustrate different encoding methods for various transformers. In Figure 1a, an original colon polyp image is divided into 4 × 4 pixel blocks. Pixel block 1 represents information about the outer side of the colon and has a relatively far distance from pixel block 2, resulting in significant differences in the represented information. Pixel block 2, being adjacent to pixel block 3, provides more information about polyps and exhibits a strong correlation. When taken as the center, pixel block 2 shows a stronger correlation with closer blocks and a weaker correlation with blocks farther away. On the right side of Figure 1, three different encoding methods for transformers are displayed, including the vision transformer, Swin transformer [34,35], and GC-VIT [36]. In these representations, different colors are used to denote different stages of the encoding process. Gray represents the uncoded patch, green represents the central patch, yellow represents the current patch that needs encoding and interacts with the central patch, blue represents the encoded patch, and purple represents the more important patches extracted by GTG (global token generation).

The vision transformer directly utilizes the self-attention mechanism to uniformly encode each patch block, ignoring the relationships between pixel blocks. The U-Net based on Swin the transformer uses the shift window to encode the central pixel block with additional neighboring pixel blocks. However, this approach can lead to resource wastage due to window sliding and the computation of masks. To address these weaknesses, the proposed GC-VIT employs GTG (global token generation) to calculate the correlations between each pixel block. This process allows for the extraction of useful pixel blocks while effectively removing irrelevant and redundant information. Instead of incorporating all information, the approach selectively includes only the useful information in the computation of global self-attention. This method does not require any additional operations, resulting in resource savings.

We compare the Swin transformer with the GC-VIT and discuss their performance. In the paper “Global Context Vision Transformers”, the authors conducted experiments on the ADE20K dataset to compare the GC-VIT with several mainstream models, including the Swin transformer. In the experiments, Swin-T has a higher number of parameters (Param (M)) compared to the GC VIT-T, while their computational complexity (FLOPs (G)) is approximately the same. Swin-B has a higher number of parameters (Param (M)) than GC VIT-T, but its computational complexity (FLOPs (G)) is significantly lower than GC VIT-B.

Based on these experimental results, it can be observed that the GC-VIT outperforms the Swin transformer significantly, indicating that the GC-VIT is more efficient than the sliding window mechanism used in the Swin transformer in terms of computational cost. The GC-VIT avoids the computational overhead associated with masking and window shifting, which is required by the sliding window approach. The sliding window mechanism often involves extensive masking and window shifting operations, especially when dealing with high-resolution images, leading to a notable increase in computation and inference time.

## 3. The Proposed Algorithm

### 3.1. The Framework of GC-TransUnet

The framework of GC-TransUnet proposed in this paper is presented in Figure 2. GC-TransUnet consists of three main components: an (1) encoder, (2) a decoder, and (3) skip connections. In the encoder section, the original image is initially divided into four patch blocks, serving as input for the four-layer GC VIT module. After encoding, the image size is reduced to (H/32) × (W/32). Subsequently, patch expansion and ConvMore operations are employed to resize the image to (H/16) × (W/16). Detailed feature layers of varying sizes are extracted using a CNN. These features are then fused with the corresponding layers in the decoder section via skip connections. The decoder achieves its functionality through upsampling and convolution operations. Further details are illustrated below.

### 3.2. Encoder

#### 3.2.1. GC-VIT

The GC-VIT module comprises four components: (1) local attention, (2) global attention, (3) a global token generator (GTG), and (4) the downsample. The local attention performs local self-attention calculations, while the global attention handles global self-attention calculations. Each self-attention mechanism consists of multihead self-attention (MSA) and multilayer perceptron (MLP) layers. The GTG component contributes global information for overall computations. Figure 3 presents the constituent elements of the GC-VIT module. Figure 3a illustrates the framework of the GC-VIT, Figure 3b showcases the GC-VIT block, and Figure 3c displays the downsample module.

(1)Local attention

The local attention component of the GC-VIT comprises local MSA and MLP layers. Local attention is inspired by the Swin transformer block. It involves dividing the image into multiple windows of equal size, where each patch block attends to other patch blocks within the same window. This approach enables efficient self-attention computation within each window.

(2)Global attention

The global attention component of the GC-VIT includes global MSA and MLP layers. Global features are extracted using the GTG. In our proposed algorithm, each encoding is performed only once. Self-attention computation is accomplished by the global feature’s query interacting with the local key and value, as described in Equation (Equation 1).
(1)$$Attention\left(Q,K,V\right)=Softmax\left(QK/\sqrt{d}+b\right)V$$
where *Q*, *K*, and *V* represent the query, key, and value matrix, respectively. *d* denotes the dimension of the key, query, and value. *b* stands for bias.

(3)Global token generator

In our network model, We use the global token generator (GTG) to extract local features, thus providing essential prerequisites for the subsequent global attention’s self-attention computation. The structure of the global token generator is depicted in Figure 4. Within the GTG, we employ the Fused MBConv block (as illustrated in Figure 4b) and the Max Pool module (maximum pooling) to transform the extracted features into sequences. This conversion facilitates subsequent global attention. The quantity of modules employed relies on the dimensions of the image and window. The reshape operation is utilized to convert the image information into sequence information.

(4)Downsample

The downsample layer is employed to amplify the channel count. In contrast to downsampling layers in CNNs, we integrate Fused-MBConv into our model. The Fused-MBConv module contains DWConv${}_{3\times 3}$, Gelu, SE, and Conv${}_{1\times 1}$. DWConv${}_{3\times 3}$ is utilized to extract features. The Gelu layer is used to normalize the features, and the SE module enhances the features at the channel level to improve the encoding of the network. Conv${}_{1\times 1}$ is to increase the channel size by a factor of 2 to adapt to subsequent encoding operations. For the downsampling operator, a convolutional layer with a kernel size of 3 and a stride of 2 is utilized. Additionally, layer normalization (LN) is connected to achieve the downsampling operation. The Fused-MBConv block is defined as Equation (Equation 2).
$$X_{1}={\tilde{DWConv}}_{3}\left(x\right)$$
$$X_{2}=\tilde{GELU}\left(X_{1}\right)$$
$$X_{3}=\tilde{SE}\left(X_{2}\right)$$
(2)$$\bar{x}={\tilde{Conv}}_{1}\left(X_{3}\right)+x$$
where ${\tilde{DWConv}}_{3}$ represents a 3 × 3 depth-wise convolution (DW-Conv) [37,38]. $\tilde{GELU}$ represents the Gaussian Error linear unit [39]. $\tilde{SE}$ signifies the Squeeze and Excitation module [40], and ${\tilde{Conv}}_{1}$ represents a 1 × 1 convolution layer.

The enhanced downsampling technique provides a more effective solution to address the inductive bias problem. Simultaneously, as the encoding iterations increase, the receptive field is expanded, promoting enhanced feature extraction. Following each downsampling step, the image dimensions are halved, while the channel count is doubled.

#### 3.2.2. Patch Expand

After applying the GC-VIT, we employ the patch expand module to reduce the image size and channels by half. Initially, a linear layer is utilized to augment the feature size and channel count. Subsequently, the feature is reorganized, reducing its size by half and the channel count to one-fourth. The feature dimensions and channels are rearranged through addition and subtraction operations.

#### 3.2.3. Feature Extraction

In our study, we utilize a CNN for image feature extraction. We employ four convolution operations. In the first convolution operation, the convolution kernel size, stride, and padding are set to 7, 2, and 3, respectively. Subsequently, we apply the Norm layer and the ReLU layer. For the remaining three convolution operations, each layer comprises two 1 × 1 convolutional layers and one 3 × 3 convolutional layer. The 1 × 1 convolutional layer employs a convolution kernel of size 1, a stride of 1, and padding of 0. The 3 × 3 convolutional layer utilizes a convolution kernel of size 3, a stride of 1, and padding of 1. To ensure feature preservation for subsequent skip connections, we adopt the residual block structure from ResNet [41] and forward the features to the next convolution. The feature information obtained from each convolution operation is retained for the skip connections in subsequent steps.

### 3.3. Decoder

Firstly, a reshape operation is performed to ensure the input format is appropriate. The transformer processing format involves inputting a one-line sequence and outputting a one-line sequence. Decoders like U-Net operate on the image height, width, and channel format. After the reshape operation, the patch blocks are unpacked. Subsequently, the ConvMore operation is executed. This operation employs a convolution operation with a kernel size of 3 and padding of 1. Its purpose is to increase the number of channels to match the number of channels required for U-Net processing, without altering the image size. Taking Figure 1 as an example, the channel C is transformed into channel 512 while keeping the image size unchanged.

Moving into the U-Net decoding part, we utilize a combination of upsampling and convolution operations. The convolution operation employs a 3 × 3 convolution kernel with a padding of 1. The upsampling operation is achieved through an upsampling layer that doubles the image size using bilinear interpolation. Prior to each convolution operation, the features generated by the CNN are fused via skip connections to enhance the decoding efficiency.

While the transformer globally encodes information, the CNN is employed to extract detailed layer information. However, as the receptive field expands, there is a risk of losing shallow-level information. To address this issue, we incorporate skip connections into the decoder, which facilitates the fusion of shallow information with upsampled information. This approach proves to be more beneficial for image segmentation tasks as it preserves important details throughout the network.

### 3.4. GC-TransUnet Configurations

GC-TransUnet configurations are presented in Table 1.

## 4. Experimental Results and Analysis

### 4.1. Dataset

(1)ISIC 2018 dataset:

The ISIC 2018 dataset comprises images of skin cancer lesions, which are provided in PNG format. This dataset consists of 2596 images along with their corresponding standard segmentation results. Figure 5 serves as an illustrative example of the ISIC 2018 image dataset. In Figure 5a, a collection of skin cancer lesion images is presented, while Figure 5b showcases the corresponding standard segmentation results depicting the segmented lesion area.

For our experiment, we divided the dataset into three subsets: a training set, a validation set, and a test set. These subsets accounted for 60%, 20%, and 20% of the total, respectively. Additionally, we resized the images to a dimension of 224 × 224. Multiple rounds of training were conducted, and we selected the best-performing model to generate the final segmentation result.

(2)CVC-ClinicDB dataset:

The CVC-ClinicDB dataset comprises images in PNG format specifically focused on colon polyps. This dataset consists of 612 images, each with a size of 384 × 288, all derived from colonoscopy sequences. Figure 6 serves as an illustrative example of the CVC-ClinicDB image dataset. In Figure 6a, a group of source images depicting colon polyps is displayed, while Figure 6b showcases the corresponding standard segmentation results representing the segmented lesion area.

Considering the specific characteristics of the images, we further divided them into a size of 384 × 384 pixels. Additionally, we partitioned the dataset into three subsets: a training set, a test set, and a validation set, with proportions of 70%, 20%, and 10%, respectively. Multiple rounds of training were performed, repeatedly iterating the training process to achieve the optimal results.

(3)Kvasir-SEG dataset

The Kvasir-SEG dataset consists of 1000 images of colon polyps in JPEG format, accompanied by their respective labels. The image sizes in the dataset vary from 332 × 487 pixels to 1920 × 1072 pixels. Figure 7 serves as an illustrative example of the Kvasir-SEG image dataset. In Figure 7a, a collection of source images depicting colon polyps is displayed, while Figure 7b showcases the corresponding standard segmentation results representing the segmented lesion area.

To facilitate training, the images are set to a standardized size of 384 × 384 pixels. The dataset is divided into a training set, a test set, and a validation set, containing 700, 100, and 200 images, respectively. Multiple training iterations are performed, and the model with the highest score is selected to demonstrate the final results.

### 4.2. Experimental Details

The GC-TransUnet model is implemented using Python 3.7 and PyTorch 1.11.0. On the GPU side, we utilize the NVIDIA GeForce RTX 3060 Laptop GPU with 6 GB of memory. Throughout the training process, we maintain a uniform patch block size of 4. The optimizer employed is RMSprop, and the loss function utilized is dice loss.

Regarding the ISIC 2018 dataset, the learning rate is set to 0.001, and the batch size is set to 6. We train the gc-vit-tiny model and repeat the training process to obtain the best value. In our experiment, we utilize various evaluation metrics to assess the experimental results, including the midce, accuracy, recall, miou, fwavacc, and F1 score. We also conduct a comparative analysis between the U-Net series network and TransUnet using the ISIC 2018 dataset. Furthermore, in the colon polyp dataset, we compare the experimental results with classic networks, such as U-Net, resulting in favorable outcomes.

### 4.3. Experimental Results on ISIC 2018 Dataset

The experimental results of different algorithms on the ISIC 2018 dataset are presented in Table 2, with the best outcomes highlighted in bold. As shown in Table 2, GC-TransUnet and TransUnet outperform other algorithms significantly in terms of recall, miou, and F1 score. In comparison to the TransUnet model, our proposed GC-TransUnet exhibits a noticeable improvement of 1.4% and 2.3% in midce and recall, respectively, while also showing enhancements in other metrics. Hence, the GC-Transformer model demonstrates superiority over other comparative models in the identification and segmentation of skin cancer lesions, particularly for larger lesions.

In order to compare the segmentation results of TransUnet and GC-TransUnet more clearly, an example of segmentation results of both algorithms on the ISIC 2018 dataset are depicted in Figure 8. In Figure 8, Figure 8a portrays the input images, while Figure 8b represents the corresponding ground truth of the lesion areas. Figure 8c and Figure 8d display the segmentation results achieved by TransUnet and GC-TransUnet, respectively. Figure 8 clearly demonstrates that, compared to TransUnet, GC-TransUnet produces more comprehensive segmentation of the lesion area, exhibiting sharper boundaries and closely resembling the ground truth at numerous prediction points. These improvements are likely attributed to the encoding of both long-range and short-range information. The results indicate the proposed model’s capability to deliver impressive outcomes in medical image segmentation.

### 4.4. Experimental Results on CVC-ClinicDB Dataset

The experimental results of different algorithms on the CVC-ClinicDB dataset are presented in Table 3, with the optimal outcomes highlighted in bold. As observed in Table 3, GC-TransUnet surpasses other models in terms of midce, accuracy, miou, and F1 score metrics. In the case of the recall index, the Attention R2U-Net model achieves the best results, with GC-TransUnet trailing by approximately 4.5%. However, GC-TransUnet outperforms Attention R2U-Net in other metrics. Notably, GC-TransUnet exhibits more than a 13% improvement in accuracy compared to Attention R2U-Net. This further emphasizes the advantage of our model in accuracy.

The segmentation results of different algorithms on the CVC-ClinicDB dataset are depicted in Figure 9. Figure 9a illustrates the input images, Figure 9b represents the corresponding ground truth of colon polyps, and Figure 9c–f depict the segmentation results obtained by U-Net, U-Net++, Attention R2U-Net, and GC-TransUnet, respectively. It is evident from Figure 9 that GC-TransUnet demonstrates a superior performance. In the case of small region segmentation (such as the input image in the first row), our model yields the most accurate predictions, while for larger region segmentation, it excels at capturing contours and protrusions within the images. Furthermore, in the second and third sets of images, our model exhibits better identification of interference caused by other pixels, thereby ensuring prediction accuracy.

### 4.5. Experimental Results on Kvasir-SEG Dataset

The experimental results of different algorithms on the Kvasir-SEG dataset are displayed in Table 4, with the best outcomes highlighted in bold. As observed in Table 4, our model outperforms other classical models across various metrics. The average midce, precision, recall, miou, and F1 score are higher by 3.49%, 4.82%, 4.96%, 4%, and 4.97%, respectively, compared to the suboptimal model (U-Net++). The experimental results demonstrate that our model achieves favorable segmentation results in colon polyp segmentation.

The segmentation results of different algorithms on the Kvasir-SEG dataset are depicted in Figure 10. In Figure 10, Figure 10a represents the input images, Figure 10b portrays the corresponding ground truth of colon polyps, and Figure 10c–f depict the segmentation results obtained by U-Net, U-Net++, ResUNet++, and GC-TransUnet, respectively. It is evident from Figure 10 that in the first set of segmentation results GC-TransUnet provides more comprehensive and accurate predictions. Moreover, in the case of small region segmentation (such as the input image in the third row), GC-TransUnet accurately predicts the shape and position of the image. In the second and fourth sets of segmentation results, GC-TransUnet demonstrates precise prediction of the segmentation area’s outline, closely resembling the ground truth in many details. Therefore, our model exhibits a superior performance in colon polyp image segmentation applications.

### 4.6. Processing Time and Computational Cost

In this section, we conducted a comparative study of different models and our proposed algorithm in terms of processing time. We used “Params (M)” to measure the number of model parameters and “FLOPs (G)” to estimate the computational complexity. We used the FPS (Frames Per Second) metric to measure the performance of each model on the same dataset. Throughout the experiments, all images were resized to a size of 224 × 224. The experimental results are presented in Table 5 below:

According to Table 5, our algorithm demonstrates a relatively low processing time, indicating high efficiency in handling images. Our algorithm outperforms Unet++ and R2AttUnet in image processing capability and slightly surpasses ResUnet++ and Att-unet, while being comparable to Unet. Although DeepLabV3+ exhibits a faster processing time, the image quality of its segmentation results is inferior to our algorithm. On the ISIC 2018 dataset, our algorithm performs more than 2% better in all evaluation metrics compared to DeepLabV3+. In the polyp dataset, our algorithm also outperforms DeepLabV3+.

We believe that achieving higher image quality is desirable even if it comes at the cost of increased processing time. By balancing both time and image quality considerations, our algorithm exhibits outstanding performance in medical image segmentation tasks. Its efficient processing speed and excellent segmentation results make it highly promising for practical applications.

And our model has a relatively small number of parameters, which is comparable to U-Net. However, in terms of computational complexity, our model requires fewer FLOPs compared to mainstream models. This indicates that our model’s inference speed will be faster. Additionally, due to its faster inference speed and appropriate parameter size, our model provides valuable assistance for future algorithm improvements.

### 4.7. Ablation Study

To investigate the impact of various factors, we conducted ablation experiments on the ISIC 2018 dataset. Specifically, we examined the effect of the distance between each patch block on encoding quality and the patch expand module in our model.

In order to investigate the influence of the distance between each patch block on encoding quality, we conducted three sets of experiments: (1) The first set of experiments involved removing global attention and replacing it with local attention, aiming to explore the significance of distant patch blocks in the encoding process. (2) The second set of experiments focused on removing the feature extraction mechanism of the GTG module while retaining other components, with the aim of exploring the necessity of feature extraction and filtering out redundant information. (3) The third set of experiments aims to investigate the impact of our proposed patch expand module compared to the traditional bilinear interpolation on image segmentation results.

(1)Effect of local attention

In order to explore the role of local attention in the entire algorithm, we conducted an ablation study on the ISIC 2018 dataset by removing local attention from the encoder. The experimental results are shown in Table 6. From Table 6, it can be observed that after disabling local attention the model’s performance decreased by 0.43% in midce, 0.61% in miou, and 0.53% in F1 score. The largest decrease was observed in accuracy, reaching 1.32%. However, the recall remained unchanged compared to the original algorithm. We believe that local attention effectively performs local encoding, leading to improved accuracy at local edge positions.

(2)Effect of global attention

By removing global attention, we intentionally eliminated the interaction between global attention and GTG modules, effectively preventing the network from encoding remote patch blocks. The impact of this modification on the encoding quality is demonstrated in Table 7. Without the presence of global attention, the transformer only encodes adjacent patch blocks, neglecting crucial information from patch blocks located further away. Consequently, this limited encoding strategy leads to suboptimal performance.

(3)Effect of GTG

In this experiment, we excluded the feature extraction mechanism from the GTG module while retaining certain pooling functionality. Without feature extraction, global attention encoded each patch block of the entire image uniformly. The experimental results are presented in Table 8. Due to the significant distance between two patch blocks, the valuable information shared between them was limited, resulting in some information becoming irrelevant or even noise. To improve encoding quality, it is crucial to filter patch blocks over longer distances, extract essential information, and mitigate the impact of noise. Hence, we incorporate a feature extraction mechanism to achieve these objectives.

(4)Effect of downsample

The most crucial part of the downsample module is the Fused-MBConv module. To understand the role of the downsample module in the algorithm, we removed the Fused-MBConv module from it and conducted experiments to analyze the impact. The experimental results are presented in Table 9. After removing the Fused-MBConv module, the overall performance of the algorithm declined. Specifically, there was a decrease of 0.65% in miou, 0.37% in accuracy, 0.78% in recall, 0.63% in F1 score, and 0.56% in midce. These results suggest that the absence of the Fused-MBConv module in the downsample module hindered the proper organization and improvement of feature maps, leading to a decline in the overall algorithm performance.

(5)Effect of patch expand module

In the proposed GC-TransUnet, we utilize the patch expand module to facilitate the necessary adjustments in image size and channels, ensuring compatibility with the decoder. In this experiment, we investigate the impact of patch expand module compared to the traditional bilinear-interpolation-based upsampling on image segmentation results. The experimental results in Table 10 demonstrate the superiority of our proposed patch expand module in achieving improved segmentation outcomes within GC-TransUnet.

## 5. Conclusions

In this paper, we introduce GC-TransUnet, an approach based on an innovative global context vision transformer for medical image segmentation. The encoder incorporates a pyramid structure inspired by the GC-VIT to enhance operational efficiency. In the decoder part, we utilize upsampling layers and convolutional layers from U-Net. To address the interaction issue, we employ both local self-attention and global self-attention mechanisms. Additionally, we use the patch expand module to effectively adjust image size information and channels.

We evaluate the performance of GC-TransUnet on three medical image datasets: ISIC 2018, CVC-ClinicDB, and Kvasir-SEG. Experimental results demonstrate the effectiveness of GC-TransUnet, as it achieves highly satisfactory segmentation outcomes. When compared to other methods, our proposed algorithm exhibits superior performance, both subjectively and objectively. For future work, we intend to explore the incorporation of the transformer into the decoder to further enhance the results.

## Figures and Tables

**Figure 1 entropy-25-01169-f001:**
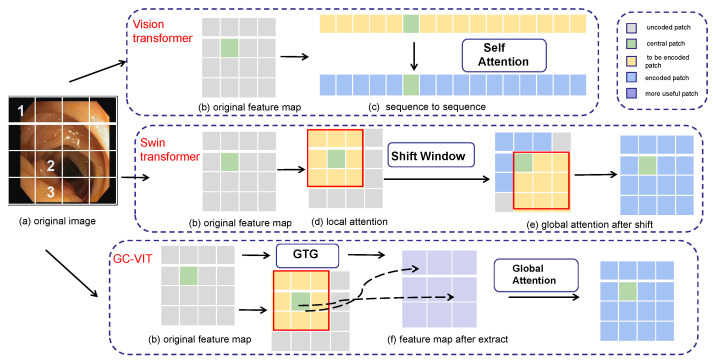
Different encoding methods for different transformers.

**Figure 2 entropy-25-01169-f002:**
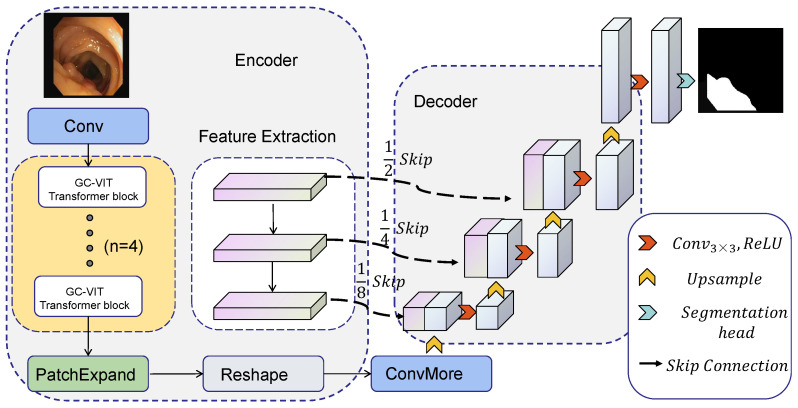
The framework of GC-TransUet.

**Figure 3 entropy-25-01169-f003:**
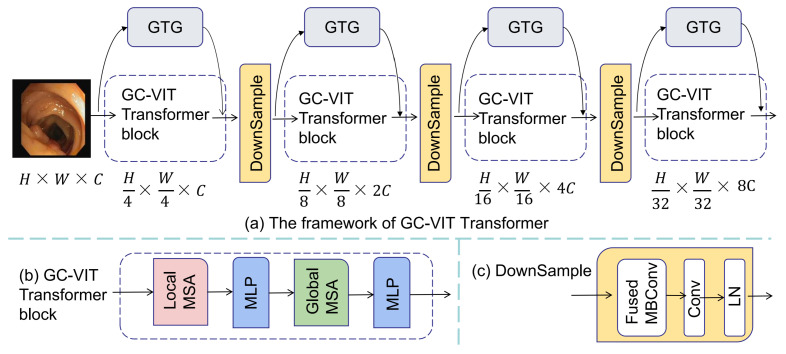
The architecture of GC-VIT.

**Figure 4 entropy-25-01169-f004:**
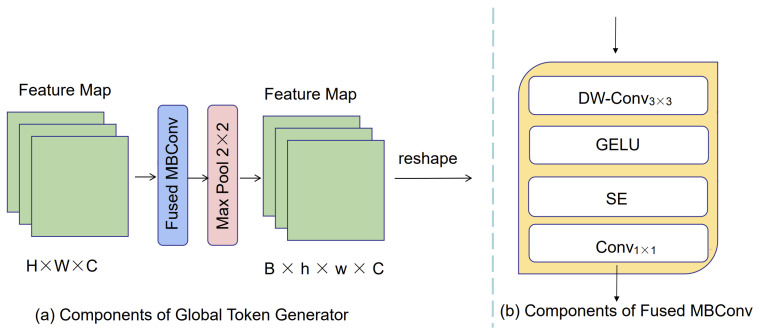
The architecture of global token generator.

**Figure 5 entropy-25-01169-f005:**
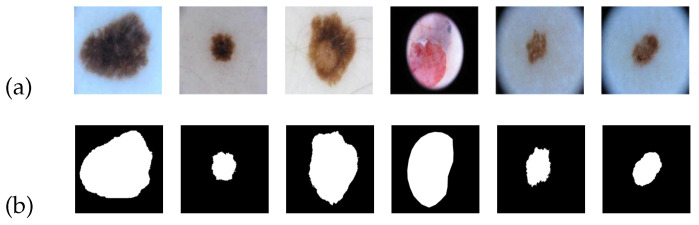
An example of ISIC 2018 dataset: (**a**) input; (**b**) ground truth.

**Figure 6 entropy-25-01169-f006:**
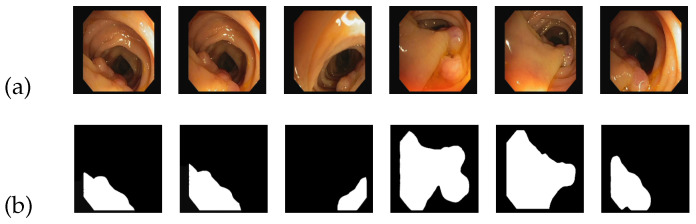
An example of CVC-ClinicDB dataset: (**a**) input; (**b**) ground truth.

**Figure 7 entropy-25-01169-f007:**
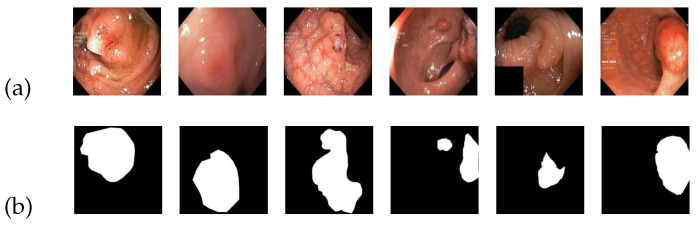
An example of Kvasir-SEG dataset: (**a**) input; (**b**) ground truth.

**Figure 8 entropy-25-01169-f008:**
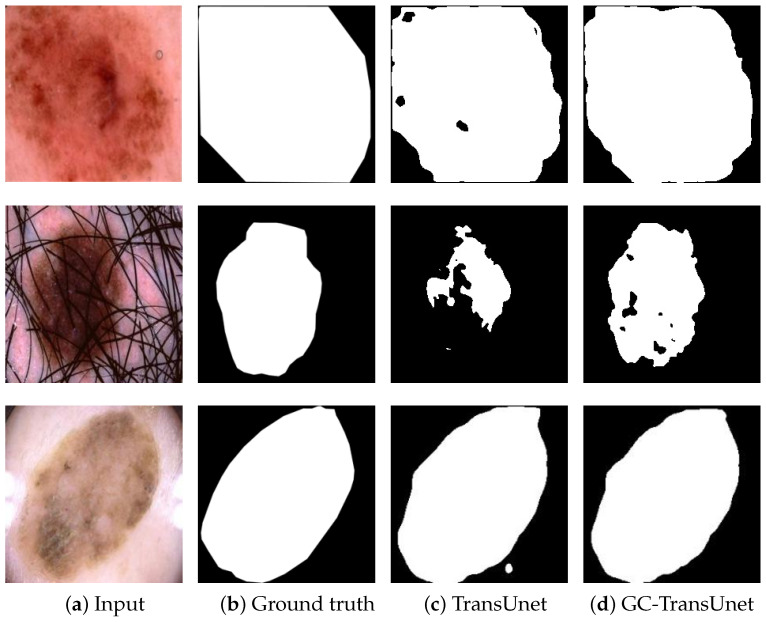
Segmentation results of TransUnet and GC-TransUnet on ISIC 2018 dataset.

**Figure 9 entropy-25-01169-f009:**
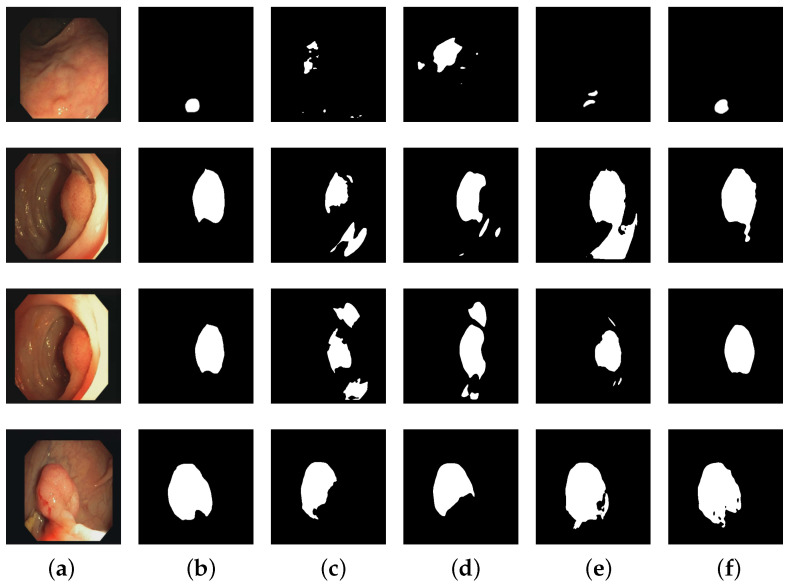
Segmentation results of different methods on CVC-ClinicDB dataset: (**a**) input; (**b**) ground truth; (**c**) U-Net; (**d**) U-Net++; (**e**) Attention R2U-Net; (**f**) GC-TransUnet.

**Figure 10 entropy-25-01169-f010:**
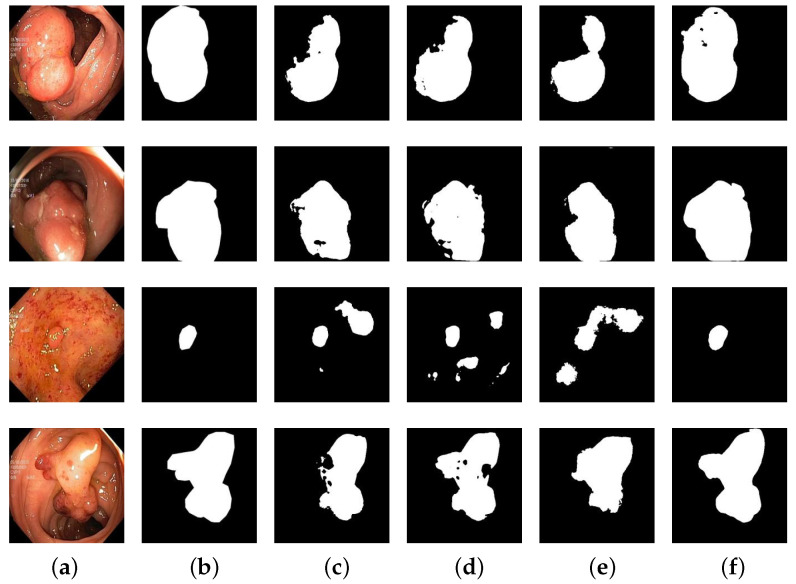
Segmentation results of different methods on Kvasir-SEG dataset: (**a**) input; (**b**) ground truth; (**c**) U-Net; (**d**) U-Net++; (**e**) ResUNet++; (**f**) GC-TransUnet.

**Table 1 entropy-25-01169-t001:** GC-TransUnet configurations.

Layer	Configurations	
	**GC VIT-T**	**GC VIT-B**
Feature extraction	Conv1 [K:7 S:2 P:3]	Conv1 [K:7 S:2 P:3]
	Group Norm [eps:10${}^{-6}$]	Group Norm [eps:10${}^{-6}$]
	ReLu	ReLu
Feature extraction Block X3	Conv2 [K:1 S:1 P:0]	Conv2 [K:1 S:1 P:0]
	Group Norm [eps:10${}^{-6}$]	Group Norm [eps:10${}^{-6}$]
	ReLu	ReLu
	Conv3 [K:3 S:1 P:1]	Conv3 [K:3 S:1 P:1]
	Group Norm[eps:10${}^{-6}$]	Group Norm [eps:10${}^{-6}$]
	ReLu	ReLu
	Conv4 [K:1 S:1 P:0]	Conv4 [K:1 S:1 P:0]
Transformer encoder	Local and global attention	Local and global attention
	Block depth [3, 4, 19, 5]	Block depth [3, 4, 19, 5]
	Num heads [2, 4, 8, 16]	Num heads [4, 8, 16, 32]
	Window size [7, 7, 14, 7]	Window size [12, 12, 24, 12]
	dim 64	dim 128
	mlp ratio 3	mlp ratio 2
Transformer encoder	GTG	GTG
	Fused MBConv	Fused MBConv
	Conv1 [K:3 S:1 P:1]	Conv1 [K:3 S:1 P:1]
	GeLu	GeLu
	SE	SE
	Conv2 [K:1 S:1 P:1]	Conv2 [K:1 S:1 P:1]
Patch expand	dim 64 × 8	dim 128 × 8
Decoder	Conv1 [K:3 P:1 BN]	Conv1 [K:3 P:1 BN]
	Conv2 [K:3 P:1 BN]	Conv2 [K:3 P:1 BN]
	Upsample	Upsample
	Bilinear interpolation	Bilinear interpolation

K represents kernel size, S represents stride, and P represents padding.

**Table 2 entropy-25-01169-t002:** Experimental results of different methods on ISIC 2018 dataset.

Methods	Midce	Accuracy	Recall	Miou	F1 Score
U-Net [12]	-	-	70.80%	54.90%	67.40%
Attention U-Net [41]	-	-	71.70%	56.60%	66.50%
Attention R2U-Net [42]	-	-	72.60%	59.20%	69.10%
DeepLabV3+ [43]	82.46%	90.60%	75.39%	79.09%	82.30%
TransUnet [25]	83.98%	95.66%	75.21%	81.27%	84.21%
GC-TransUnet	**85.38%**	**95.82%**	**77.59%**	**82.82%**	**85.75%**

**Table 3 entropy-25-01169-t003:** Experimental results of different methods on CVC-ClinicDB dataset.

Methods	Midce	Accuracy	Recall	Miou	F1 Score
DeepLabV3+ [44]	67.95%	51.85%	55.34%	63.11%	53.53%
Attention U-Net [42]	71.16%	69.90%	52.81%	67.77%	60.16%
U-Net [12]	72.76%	75.09%	54.53%	69.68%	63.18%
U-Net++ [45]	75.29%	71.63%	62.21%	71.58%	66.59%
Attention R2U-Net [43]	77.76%	71.62%	**68.46%**	73.73%	70.00%
GC-TransUnet	**78.61%**	**84.87%**	63.92%	**76.11%**	**72.92%**

**Table 4 entropy-25-01169-t004:** Experimental results of different methods on Kvasir-SEG dataset.

Methods	Midce	Accuracy	Recall	Miou	F1 Score
DeepLabV3+ [44]	67.63%	67.77%	50.83%	63.24%	58.09%
ResUNet++ [46]	70.97%	71.58%	56.12%	66.80%	62.92%
Attention U-Net [42]	74.50%	78.78%	60.57%	70.82%	68.49%
U-Net [12]	76.43%	84.85%	61.96%	73.28%	71.62%
U-Net++ [45]	77.79%	82.89%	65.76%	74.43%	73.33%
GC-TransUnet	**81.28%**	**87.71%**	**70.72%**	**78.43%**	**78.30%**

**Table 5 entropy-25-01169-t005:** Processing time and computational cost of each model.

Model	FPS	Params (M)	FLOPs (G)
R2AttUnet	5.81	39.44	117.98
Unet++	6.36	47.18	153.21
ResUnet++	13.05	14.48	54.35
Att-unet	16.15	34.87	51.01
Unet	18.40	34.52	50.16
Ours	17.26	33.13	11.72
DeepLabV3+	31.30	21.54	34.97

**Table 6 entropy-25-01169-t006:** Experimental results of local attention on the ISIC 2018 dataset.

Methods	Midce	Accuracy	Recall	Miou	F1 Score
Not local attention	84.95%	94.50%	77.59%	82.21%	85.22%
GC-TransUnet	85.38%	95.82%	77.59%	82.82%	85.75%

**Table 7 entropy-25-01169-t007:** Experimental results of global attention on the ISIC 2018 dataset.

Methods	Midce	Accuracy	Recall	Miou	F1 Score
Not global attention	83.04%	94.96%	73.91%	80.16%	83.12%
GC-TransUnet	85.38%	95.82%	77.59%	82.82%	85.75%

**Table 8 entropy-25-01169-t008:** Experimental results of GTG on the ISIC 2018 dataset.

Methods	Midce	Accuracy	Recall	Miou	F1 Score
Not GTG	84.20%	94.67%	76.14%	81.40%	84.40%
GC-TransUnet	85.38%	95.82%	77.59%	82.82%	85.75%

**Table 9 entropy-25-01169-t009:** Experimental results of downsample on the ISIC 2018 dataset.

Methods	Midce	Accuracy	Recall	Miou	F1 Score
Not downsample	84.82%	95.45%	76.81%	82.17%	85.12%
GC-TransUnet	85.38%	95.82%	77.59%	82.82%	85.75%

**Table 10 entropy-25-01169-t010:** Experimental results of patch expand module on the ISIC 2018 dataset.

Methods	Midce	Accuracy	Recall	Miou	F1 Score
Bilinear interpolation	83.33%	94.54%	74.67%	80.44%	83.44%
Patch expand	85.38%	95.82%	77.59%	82.82%	85.75%

## Data Availability

Our data presented in this study are openly available in ISIC 2018 Challenge—Task 1: Lesion Boundary Segmentation at https://challenge.isic-archive.com/landing/2018/45/ (accessed on 7 June 2022), Kvasir-SEG at https://datasets.simula.no/kvasir-seg/#download (accessed on 6 October 2022), and CVC-ClinicDB at https://paperswithcode.com/dataset/cvc-clinicdb (accessed on 20 August 2022).

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
