# Peer review of "Distance Matters: A Distance-Aware Medical Image Segmentation Algorithm"

_entropy, 2023, doi:10.3390/e25081169_

Round 1
Reviewer 1 Report
This paper proposed a novel GC-TransUnet for medical image segmentation that considers the effect of different scales of long-range information during the coding process, thereby reducing resource consumption. Experimental results show superior results compared to other algorithms. However, some general issues should be addressed before it fits the standard to be published.
1. The Abstract should be improved so that the main contribution of this work can be better understood by the readers.
2. Please include a layer-wise configuration detail of the proposed method in a separate table.
3. There are a lot of state-of-the-art deep segmentation models (such as DeepLabV3+). Please include the comparative results of DeepLAbV3+.
4. In the proposed GC-TransUnet, how local attention and global attention part contribute in performance gain in the proposed model?
5. The authors ignore the discussion of the efficiency of their proposed and different baseline models in terms of processing time (i.e., FPS).
6. Add some recent literature related to classification and segmentation of medical imaging
for example
>>Accurate segmentation of nuclear regions with multi-organ histopathology images using artificial intelligence for cancer diagnosis in personalized medicine
>>Automated Diagnosis of Various Gastrointestinal Lesions Using a Deep Learning–Based Classification and Retrieval Framework With a Large Endoscopic Database: Model Development and Validation
>>MDFU-Net: Multiscale dilated features up-sampling network for accurate segmentation of tumor from heterogeneous brain data
>>Volumetric Model Genesis in Medical Domain for the Analysis of Multimodality 2D/3D Data based on the Aggregation of Multilevel Features
>>Artificial Intelligence-based computer-aided diagnosis of glaucoma using retinal fundus images
7. It is highly recommended to make the proposed model publicly available through GitHub or other sources.
1. Please perform the English revision of this paper by a native speaker.
Reviewer 2 Report
The proposed method in this manuscript is to design a new way on segmentation Transformer model. In order to overcoming demerits of the vanilla and Swin Transformer models, the authors proposed a Global-Context ViT. In GC-ViT model includes more efficient local and global feature extractions and is combined with U-Net-based decoder to generate segmentation mask. The results shows its outperformance, however, there are some points to be addressed if it reaches the publication level.
1. It needs more details about how GC-ViT can reduce the computing power in comparison to Swin Transformer.
2. For the Global attention, why does global token generator extract global features? It seems channel-wise combinations.
3. Due to lack of intuitive explanation about each module, it would be better to show ablation studies about performance validation of each module.
4. Can you apply other transformer encoder to your decoder? Please provide the computational cost as well as accuracy.
5. Please add SOTA U-Net based segmentation such that "Seo, Hyunseok, et al. "Modified U-Net (mU-Net) with incorporation of object-dependent high level features for improved liver and liver-tumor segmentation in CT images." IEEE transactions on medical imaging 39.5 (2019): 1316-1325."
6. There is some typos. For example, in Figure 4 caption, toke -> token
Round 2
Reviewer 1 Report
The authors have addressed all the comments correctly. No more comments.
Please proof read the final version before publication.
Reviewer 2 Report
Thanks much for your effort in this revision round. Most of my concerns about the manuscript have been addressed. I have one more quick question.
Can you apply any pre-trained parameters in your transformer model? There are some pre-trained parameters for vanilla ViT and Swin transformer. So, basically, we can apply them and get an improved results by fine-tuning. So, please discuss whether the proposed model can be preferable over the pre-trained transformer model providing high performance.
